# The P2X7 Receptor: A Promising Pharmacological Target in Diabetic Retinopathy

**DOI:** 10.3390/ijms22137110

**Published:** 2021-07-01

**Authors:** Matteo Tassetto, Anna Scialdone, Anna Solini, Francesco Di Virgilio

**Affiliations:** 1Medical Residency Program in Clinical Pathology, Department of Medical Sciences, University of Ferrara, 44121 Ferrara, Italy; matteo.tassetto@edu.unife.it (M.T.); anna.scialdone@edu.unife.it (A.S.); 2Department of Surgical, Medical, Molecular and Critical Area Pathology, University of Pisa, 56126 Pisa, Italy; anna.solini@med.unipi.it; 3Department of Medical Sciences, University of Ferrara, 44121 Ferrara, Italy

**Keywords:** diabetic retinopathy, P2X7, inflammation, angiogenesis, neural damage

## Abstract

Diabetes is a worldwide emergency. Its chronic complications impose a heavy burden on patients, health systems, and on society as a whole. Diabetic retinopathy is one of the most common and serious complications of diabetes, and an established risk factor for blindness in adults. Over 15 years of investigation led to the identification of vascular endothelial growth factor (VEGF) as a main pathogenic factor in diabetic retinopathy and to the introduction of highly effective anti-VEGF-based therapies, such as the monoclonal antibody bevacizumab or its fragment ranibizumab, which helped to prevent diabetes-related blindness in millions of patients. Recently, a pathogenic role for uncontrolled increases in the extracellular ATP concentration (eATP) and for overactivation of the purinergic receptor P2X7 (P2X7R) has been suggested. The P2X7R is an eATP-gated plasma membrane channel expressed in multiple tissues and organs, with a pleiotropic function in inflammation, immunity, cancer, and hormone and growth factor release. P2X7R stimulation or overexpression positively regulate the secretion and buildup of VEGF, thus promoting neo-angiogenesis in a wide variety of disease processes. In this review, we explore current evidence that supports the role of P2X7R receptor signaling in the pathogenesis of diabetic retinopathy, as well as the most appealing current therapeutical options for P2X7R targeting.

## 1. Introduction

It is estimated that over 400 million people are currently affected by diabetes mellitus worldwide, with the incidence steadily increasing year by year. Chronic complications of diabetes are of great concern for patients, doctors, and society as a whole, even from an economic and global health perspective. Diabetic retinopathy is one of the most common and serious complications of diabetes, and a highly impactful risk factor for blindness in adults. The pathogenic role of VEGF in diabetic retinopathy is well established. In fact, uncontrolled growth of blood vessels stimulated by VEGF underlies many pathological processes, diabetic retinopathy included. This body of knowledge, accumulated over more than 15 years of pre-clinical and clinical investigation, finally led to the introduction in the clinic of nowadays consolidated treatments based on anti-VEGF monoclonal antibodies or fragments thereof, such as bevacizumab or ranibizumab, respectively, which helped to prevent diabetes-related blindness in millions of patients. Recently, a strong link has been highlighted between VEGF-dependent neo-angiogenesis and the P2X7 purinergic receptor (P2X7R). The P2X7R is a member of the P2XR family of ATP-gated plasma membrane receptors expressed in multiple cell types and tissues, with a pleiotropic function in the regulation of inflammation, immunity, and hormone and growth factors release. P2X7R stimulation or overexpression trigger secretion of VEGF and promote angiogenesis in a variety of pathophysiological conditions, cancer and diabetic retinopathy included. This has prompted studies aimed at testing the potential therapeutic applications of P2X7R targeting to prevent pathological angiogenesis. Preclinical data in selected degenerative retinopathies, and specifically in diabetes-associated retinal degeneration, are very encouraging.

## 2. Diabetes Mellitus

The word “diabetes” defines a diverse array of chronic metabolic conditions, all characterized by hyperglycemia in absence of treatment, resulting from deficient insulin secretion, insulin action, or both. Chronic effects of hyperglycemia in patients with diabetes are associated with organ damage, leading to severe and sometimes irreversible complications, affecting especially the eyes, kidneys, peripheral nerves, and the cardiovascular system [1]. The worldwide incidence of diabetes is constantly increasing, with India, China, US, and Italy all in the top 10 positions. The International Diabetes Federation predicts that the number of people living with diabetes will rise from 463 million in 2019 (1 in 11 adults) to 700 million by 2045. The number of patients affected by diabetes in Western countries has almost doubled in the last decades, in parallel with the increase rate of obesity and metabolic syndrome in the general population [2]. The worldwide prevalence of diabetes was estimated to be 2.8% in 2000 with a projection to rise to 4.4% in 2030 [3]. Diabetes is conventionally portioned into two main clinical entities: type 1 diabetes mellitus (T1D) and type 2 diabetes mellitus (T2D). In T1D, hyperglycemia is caused by an immune-mediated destruction of pancreatic β-cells leading to an absolute deficiency of insulin production, whereas the pathogenetic mechanism underlying T2D involves a complex interplay between insulin-resistance and abnormalities in β-cells function, often due to pathologic obesity and dyslipidemia, and to other causes and factors, such as low-grade chronic systemic inflammation [4]. Most of the time, patients go to their doctor not because of hyperglycemia, which unless discovered by regular check-ups or screening is in itself asymptomatic, but because of the clinical manifestation of systemic or organ-specific complications caused by their hyperglycemic state. These complications can be acute, representing a medical emergency needing immediate treatment, or chronic, with a slower progression and a major impact on long-term quality of life. Diabetic ketoacidosis (DKA) and hyperglycemic hyperosmolar state (HHS) are the main acute complications which are by themselves life-threatening and must be treated with quick fluid resuscitation and insulin administration to normalize blood glucose levels and counteract metabolic acidosis [5].

DKA and insulin-induced severe hypoglycemia are more common in T1D, while HHS without ketoacidosis, which can progress to coma if untreated, is more frequently seen in patients with T2D [6]. Chronic complications are conversely a long-term issue, responsible for the clinical, social, and economic burden associated with diabetes in older patients. Historically, the classification of chronic complications of diabetes encompassed two main categories based on the type of blood vessels damaged by hyperglycemia: macrovascular and microvascular. Those categories, even if a little outdated, are still broadly used in clinical settings. In particular, macrovascular complications include coronary heart disease, ischemic stroke, and atherosclerotic peripheral vascular disease, while microvascular complications take into account renal, ocular, and neurological dysfunctions, leading respectively to end-stage renal disease (ESRD) and retinopathy, which has a high risk of blindness which often lead patients to seek medical care, and peripheral neuropathy. This latter can impair pain sensing, thus predisposing patients to the development of lower limb gangrene, which often goes unrecognized and may require amputation [7].

### Diabetic Retinopathy

Diabetic retinopathy (DR) is one of the most common and serious chronic complications of diabetes, affecting around 47–70% of patients with a diagnosis of T1D, which counts for 10-20% of all diabetes cases, and 26–40% of patients with T2D [8,9]. Almost the totality of type 1 diabetes patients will develop some degree of retinal damage, which will also affect in the same timeframe more than 60% of patients with type 2 diabetes [10]. A 2013 report from the American Academy of Ophthalmology estimated a global prevalence of diabetic retinopathy at around 93 million people [11]. Patients with early diabetic retinopathy commonly have retinal asymptomatic microaneurysms, reduced perfusion, and capillary degeneration, later followed by ischemic hypoxia that in turn stimulates vasoproliferative factors such as VEGF. This slow degenerative process favors the progression towards the most severe and late forms of proliferative diabetic retinopathy (PDR) that, if left untreated, can progress to diabetic macular edema (DME), occurring in up to 12% of patients with T2D, and blindness, occurring in about 50% of untreated patients within 5 years of the diagnosis [12,13]. Chronic hyperglycemia, hypoxia, and release of inflammatory molecules (e.g., tumor necrosis factor, TNF-α, or Interleukin-1β, IL-1β) have a strong impact on retinal homeostasis by promoting on the one hand vascular dysfunction and pathological angiogenesis, both a cause of proliferative diabetic retinopathy, and on the other increased capillary permeability, with the associated production of hemorrhages and inflammatory exudate [14]. These processes in the final stage of the disease lead to legal blindness, with a 2.4-fold increased risk in patients with diabetes compared to an individual without diabetes [15]. The World Health Organization estimates that 2.6% of vision impairment and blindness can be assigned to diabetic retinopathy [16]. Blindness is of course a cause of substantial individual discomfort and a heavy burden for families and society, making diabetes the main risk factor for preventable blindness in the working-age population between 20 and 74 years in developed countries [17]. Treatment of diabetic retinopathy includes a strict metabolic control combined with anti-angiogenetic drugs and corticosteroids, but it is not very effective, as it is estimated that the proliferative forms of the disease still appear after 30 years of disease in about 20% of patients with T2D treated with intensive metabolic control [18]. Therefore, an in-depth investigation of the cellular and molecular mechanisms underlying diabetic retinopathy is of vital importance for the development of new and more efficient treatment options.

## 3. Pathogenesis of Diabetic Retinopathy

### 3.1. Neovascularization

Vascular lesions are one of the hallmarks of diabetic retinopathy and vascular endothelial growth factor (VEGF), a central player in the onset and progression of vascular damage in the retina. The key role of VEGF has been confirmed by the protective action provided by anti-VEGF drugs used to preserve visual acuity in long-term diabetic patients [18] (Figure 1). Although the VEGF family is composed of different members, the VEGF-A isoform plays the main role in the process of angiogenesis both in physiological and pathological conditions [19]. In physiological conditions, autocrine secretion of VEGF by endothelial cells and binding to its cognate receptor VEGF-R2 is required for the homeostasis of blood vessels in adults [20]. VEGF, besides its essential function in the maintenance of vascular homeostasis in healthy tissues, is also involved in the pathogenesis of a large number of diseases, including cancer, where it promotes tumor growth, invasiveness, and metastasis, and eye disease such as age-related macular degeneration (AMD), diabetic retinopathy and hypertension. It is in fact a common finding that uncontrolled growth of blood vessels promotes or facilitates numerous disease processes, including tumors and intraocular vascular disorders such as diabetic retinopathy [21,22]. VEGF can be also stored and released by glial cells (Müller cells) of the retina, and it is well documented that pharmacological inhibition of VEGF release from Müller cells decreases nuclear factor-k-light-chain-enhancer of activated B cells (NF-κB) activation and TNFα expression in diabetic mice, thus mitigating retinal inflammation and vascular leakage [23].

### 3.2. Inflammation

Inflammation is a complex homeostatic response to endogenous or exogenous harmful agents involving the combined action of effector cells and molecular mediators, culminating in complex changes in the perfusion and permeability of the microcirculation, and in the recruitment and activation of specialized cell types. As any homeostatic response, inflammation is beneficial when the perturbing stimulus is quickly removed by the concerted action of soluble factors (inflammatory mediators) and inflammatory cells, and therefore, the status quo ante is re-established, with little residual or even no anatomo-functional damage (“restitutio ad integrum”). On the contrary, persistence of the triggering agent, or dysregulation of intrinsic feed-back regulatory mechanisms of inflammation, may cause a prolonged response (chronic inflammation) with potentially serious consequences [14]. An involvement of inflammation in diabetic retinopathy was hypothesized as far back as 1964 by Powell and Field, based on the observation that patients administered with aspirin for rheumatoid arthritis showed a lower incidence of diabetic retinopathy [24]. This early finding was followed by several additional studies showing that subclinical retinal inflammation caused by activated microglia or activated macrophages and local release of inflammatory mediators are a major factor in the pathogenesis of foundational features of diabetic retinopathy, such as retina infiltration by leukocytes, breakdown of the blood–retinal barrier (BRB), vascular leakage, and neoangiogenesis [25,26]. The hypothesis that inflammation has a central role in diabetic retinopathy received a strong support by the work of Chaurasia and colleagues, showing that Akimba mice, a well-established model of diabetes, have increased intra-retina levels of the inflammasome constituents NOD-like receptor pyrine-containing 3 (NLRP3), apoptosis-associated speck-like protein containing a CARD (ASC), and caspase-1, as well as IL-1β and pro-angiogenic markers, such as VEGF, and intercellular adhesion molecule 1 (ICAM-1) compared to WT mice [26]. Additional studies have recently confirmed the key role of inflammation and the associated increased vascular permeability and exudate formation in diabetic macular edema, as well as the contribution of vascular obstructions and angiogenesis to the development of proliferative diabetic retinopathy [27].

### 3.3. Neuronal Damage

Along with microcirculatory impairment and inflammation, there is evidence pointing to early pathologic changes also occurring in the neural retina as a hallmark of diabetic retinopathy, with apoptosis mainly affecting retinal ganglion cells (RGCs) and photoreceptors, and reactive gliosis involving Müller cells and astrocytes [28]. Retinal neurodegeneration might also depend on the activation of the inflammatory cells of the retina microglia, which in some cases precedes the vascular abnormalities, embodying an early and vascular-independent contribution to the development of diabetic retinopathy [29]. Injury to neural cells causes thinning of the RGC layer and of the retinal fiber layer. However, in some patients, the retina can also appear thicker at the optical coherence tomography (OCT), due to the inflammatory edema and gliosis [29]. Retinal neurodegeneration in T2D is also frequently associated with peripheral neuropathy, contributing to establish a multi-faceted neurovascular clinical syndrome [30]. It is, therefore, appropriate to consider diabetic retinopathy as a complex pathology of the “neurovascular unit” as a whole integrated system, whose progressive disintegration is precipitated by chronic high glucose levels and inflammation, which may manifest itself as clinically manifest visual impairment in patients with chronic diabetes [31].

### 3.4. Puringergic Signaling

Adenosine triphosphate (ATP) is a purine nucleotide with a key role in intracellular energy metabolism as well as an extracellular messenger involved in a multiplicity of intercellular communication pathways in a variety of pathophysiological conditions in the cardiovascular, endocrine, immune, central, and peripheral nervous systems, and in inflammation and cancer [32]. Purinergic signaling is mediated by extracellular ATP (eATP) and its metabolites ADP and adenosine acting at two specific receptor families named P1, if activated by adenosine, or P2, if activated by ATP or ADP (but also by UTP, UDP, or UDP-glucose). P1 receptors include the A1, A2A, A2B, and A3 subtypes, while P2 receptors are further subdivided into the metabotropic, G-protein-coupled, P2Y, and the ionotropic, intrinsic ion channels, P2X. P2Y receptors (P2YR) and P2X receptors (P2XR) contain 8 and 7 members, respectively. While different nucleotides (ATP, ADP, UTP, UDP, and UDPglucose) are agonists at the P2YRs, the only physiological ligand at the P2XRs is ATP [33]. In the P2XR subfamily, P2X7R stands out for its peculiar features, since this receptor can function both as a cation-permeant ion channel and a non-selective pore. The P2X7R is a homo-trimeric receptor formed by the assembly of three P2X7 subunits that, with a mechanism still poorly understood, in the presence of sub-millimolar eATP concentrations can generate a large plasma membrane pore (often referred to as “macropore”), enabling passage of molecules of MW up to 900 Da [34]. The basic subunit of the P2X7R (the P2X7 subunit) is a transmembrane protein with a large extracellular domain and the N- and C-termini both on the cytoplasmic side [35]. The ectodomain harbors three ATP binding sites, and this nucleotide is the only known physiological ligand. Interestingly, several negative or positive allosteric modulators of the P2X7R have been described [36,37]. The pathophysiological significance of the dual functional activity of the P2X7R as an ion channel and as a non-selective pore is unknown, but it is hypothesized that the channel activity and the pore function are both important and support different P2X7R-associated responses. In fact, P2X7R stimulation promotes a wide range of cellular responses, ranging from proliferation to cell death, from cytokine release to reactive oxygen species (ROS) production. A hint to the contribution of the channel or the macropore to specific cellular responses is provided by the presence of naturally occurring variants of the human P2X7 subunit lacking a large section of the carboxyl-terminal domain, e.g., the variant known as P2X7B. This truncated variant lacks cytotoxicity but retains the growth-stimulating activity [38]. The P2X7R has a special place in inflammation since it is the most potent stimulant of the NLRP3 inflammasome and of mature IL-1β and IL-18 release, and therefore, it is a key player in antigen presentation and in the overall process of stimulation of innate and adaptive immunity [34]. The P2X7R is mainly, albeit not exclusively, expressed by immune cells, i.e., dendritic cells, macrophages, polymorphonuclear granulocytes, and T and B lymphocytes, where it fulfills different tasks, e.g., promotion of growth, differentiation, cytokine, or growth factor release, depending on the specific cell type. Since it is now an established fact that eATP accumulates to a concentration of tens or even hundreds of micromoles/L at sites of inflammation, it is obvious that purinergic signaling is a key factor in regulating inflammatory states in different organs and tissues, including endocrinopathies such as diabetes and its complications [32]. In this regard, it is noticeable that purinergic receptors are widely expressed by endocrine glands, for example the pancreas, where eATP stimulates insulin secretion. Expression of P2X7R in β-cells of pancreatic islets is substantially increased in a model of streptozotocin-induced diabetes [39]. In the chronic low-grade inflammation state characterizing T2D, it can be hypothesized that overexpression of the P2X7R sensitizes pancreatic cells to the elevated levels of eATP, or that the local eATP increase stimulates, via the P2X7R, the NLRP3 inflammasome of infiltrating inflammatory cells to secrete mature IL-1β, and thus, generate a self-amplifying, diabetes-promoting, inflammatory loop in the pancreas [40]. Additionally, IL-1β accelerates the development of insulin resistance and neurovascular damage, both of which are very relevant in the development of diabetic retinopathy [41]. Although it is still debated whether the impairment of β-cell function depends on a direct damage induced by an eATP excess or on the pro-inflammatory activity of eATP-mediated stimulation of the P2X7R, it is very likely that both processes are involved [42]. The NLRP3 inflammasome can be considered a transducer that responds to a P2X7R-triggered decrease in cytoplasmic K^+^ by promoting the release of pro-inflammatory chemical mediators, thus generating an integrated link between inflammation and purinergic signaling [43]. Angiogenesis and purinergic signaling are also closely intertwined as stimulation of the P2Y1R, known to be expressed by endothelial cells, transactivates VEGF-receptor 2 (VEGFR2), whereas stimulation of the P2X7R drives VEGF release [44]. The elevated eATP levels found at inflammatory sites may promote P2X7R-mediated hypoxia-inducible factor-1α (HIF-1α) activation in endothelial and innate immune cells. HIF-1α, the most important transcription factor stimulated by ischemic hypoxia, in turn drives VEGF release and abnormal vascular proliferation [45].

## 4. Role of the P2X7R in Diabetic Retinopathy and Its Potential Role as a Therapeutic Target

### 4.1. The P2X7 Receptor as a Target to Restore the Blood–Retinal Barrier and Reduce Inflammation

Many cell components of the retina, retinal pigmented epithelium, and possibly neuronal cells express the P2X7R, and accordingly, several retinal functions are either affected by pharmacological P2X7R blockade or impaired by genetic P2X7R deletion [46]. Converging evidence supports a role for the P2X7R in the breakdown of the blood–retinal barrier (BRB), in pericyte dysfunction, and in retinal neuronal damage in diabetes [47]. The BRB is a well-known defensive structure made of endothelial cells, microglia, astrocytes, and pericytes that protects the retina against high glucose-induced damage. The integrity of the BRB is maintained by the concerted action of several components, among which tight junctions play a fundamental role. Increased levels of VEGF-A mainly secreted by astrocytes might impair the BRB integrity by targeting claudin-5 and zonula occludens-1 (ZO-1), the two main tight junction isoforms in the retina [48]. Loss of pericytes is an early finding in diabetic retinopathy, and a factor that dramatically alters BRB permeability. Several components of the BRB may benefit by P2X7R targeting. Platania and co-workers showed that P2X7R allosteric inhibitors abrogated the in vitro damaging effects of high glucose in human retinal pericytes and prevented the acquisition of an inflammatory phenotype. This was one of the first reports demonstrating that IL-1β release from retinal pericytes exposed to high glucose is a P2X7R-dependent process, thus suggesting a novel potential pharmacological target for the treatment of diabetic retinopathy [49]. Overexpression of IL-1β and TNF-α in retinal microglia and macrophages, and of CD40 ligand, a member of the TNF-α superfamily, in Müller cells, endothelial cell, and microglia is an early feature of diabetic retinopathy and the main factor responsible for retinal inflammation in experimental models of diabetic retinopathy [50]. CD40-activated Müller cells secrete eATP via a phospholipase Cγ (PLCγ)-mediated mechanism and trigger P2X7R-mediated cytokine release from monocytes/macrophages [51]. Blood levels of the CD154 cofactor and expression of CD40 in Müller glia and P2X7R in the retinal endothelial cells and microglia/macrophages are increased in diabetes, increasing the pathogenic effect of the CD40-ATP-P2X7R axis in these patients [52] (Figure 2).

The hypothesis that high glucose levels found in diabetes increase the susceptibility to injury of retinal blood vessels via a P2X7R-mediated mechanism is supported by studies showing that even low concentrations of P2X7R agonists, usually unable to trigger cell death in the healthy retina, caused macropore formation and apoptosis in the diabetic retina [48,53,54]. In an experimental model of diabetic retinopathy in which human retinal endothelial cells were exposed to high glucose and to the synthetic P2X7R agonist BzATP, administration of the P2X7R selective antagonist JNJ47965567 rescued the main physiological markers of a healthy BRB, i.e., claudin-5 and ZO-1, and restored the trans-endothelial electrical resistance (TEER) [55]. Similarly, a natural P2X7R diterpenoid inhibitor named dihydrotanshinone (DHTS) showed a protective effect on BRB structure, very much akin to the JNJ47965567 compound [56]. In this same study, administration of DHTS prior to exposure to high glucose and BzATP prevented overexpression of both P2X7R and VEGF-A, with a strongest effect on the latter, as well as of ROS and other factors involved in the pathogenesis of diabetic retinopathy, such as ICAM-1.

### 4.2. Targeting the P2X7 Receptor to Reduce Neoangiogenesis

Another therapeutically promising pathway was hinted at by the identification of the P2X7R as an upstream modulator of the PI3K/Akt pathway in different cell types, endowed with the ability to reorganize metabolic pathways in unfavorable external conditions [57,58]. An interesting finding that might open new avenues for the treatment of diabetic retinopathy is the observation that P2X7R stimulation also caused an increase in the HIF-α expression and VEGF secretion, which were both largely reduced by the in vivo administration of the P2X7R selective antagonists A740003 and AZ10606120. Of interest, neo angiogenesis was also inhibited, reinforcing the concept already established in tumors where cell growth and neoangiogenesis in P2X7R-expressing tumors can be inhibited by targeting VEGF and/or the P2X7R [57,59]. These early findings suggest that P2X7R targeting might be an efficient avenue to inhibit VEGF release. These anticipations were confirmed by a study investigating the effect of P2X7R targeting in a model of streptozotocin-induced hyperglycemic retinopathy in rats [60]. In this model, characterized by increased expression of the P2X7R in the retina, administration of two different P2X7R antagonists, A740003 or AZ10606120, completely reverted the increased vessel permeability and reduced VEGF and IL-6 release. Interestingly, P2X7R blockade had no effects on protein leakage in the absence of streptozotocin treatment, suggesting lack of interference with the physiological regulation of vessel permeability.

### 4.3. P2X7 Receptor and Neuronal Damage

Given the expression of the P2X7R by neuronal cells in the retina, it is also conceivable that the increased levels of eATP thought to be present in the retina of diabetic patients might contribute to visual impairment by injuring P2X7R-expressing neurons, though this mechanism is as yet purely speculative because no direct measurements of eATP have ever been performed in the retina [60]. This notwithstanding, previous observations in a model of experimental glaucoma shows that degrading eATP or blocking P2X receptors prevented pressure-induced RGC damage [61]. In human organotypic retinal cultures, it was also demonstrated that the activation of the P2X7R can kill RGCs [62], a pathway that might be relevant in neuronal damage under ischemic conditions as well as in diabetic retinopathy, conditions both thought to be characterized by increased levels of eATP. One of the main pathways for non-lytic eATP release, i.e., pannexin-1 (Panx-1), might also be involved. A recent study showed that retinal ganglion cells from Panx-1-deficient animals exposed to ischemia undergo lower increase in plasma membrane permeability and reduced activation of the NLRP3 inflammasome, and are, therefore, resistant to the ischemic insult [63]. Therefore, an additional pathogenic mechanism promoted by neuro-inflammation in diabetic retinopathy might be P2X7R-activated, Panx-1-dependent RGC injury [63]. Injury- or inflammation-induced eATP release might then trigger a self-amplifying loop that will further extend neuronal damage in the retina. The pathogenic role of the P2X7R-NLRP3 inflammasome axis extends well beyond diabetic retinopathy to as of yet intractable retinal diseases such as geographical atrophy in age-related macular degeneration (AMD) [64].

An investigation of an experimental model of AMD revealed that nucleoside reverse transcriptase inhibitors (NRTIs), such as stavudine and azidothymidine, are potent blockers of P2X7R and of the associated NLRP3 inflammasome activation, and accordingly, they prevent geographic atrophy and choroidal neovascularization [64]. Recently, another NRTIs, i.e., lamivudine, was shown to inhibit the P2X7R and prevent diabetic retinopathy, thus reinforcing the role of the P2X7R and NLRP3 in its pathogenesis [65].

### 4.4. Other Potential Roles of the P2X7 Receptor as a Therapeutic Target in Diabetic Retinopathy

Chronic hyperglycemia causes the accumulation of pro-inflammatory advanced glycosylated end-products (AGE) in the retina. The P2X7R-NLRP3 axis is also activated under these conditions in human retinal microvascular endothelial cells (hRMEC). The administration of relaxins, a family of anti-inflammatory, anti-apoptotic, and anti-fibrotic compounds belonging to the insulin super-family, displayed reduced hyperglycemia-induced damage in the eye and other tissues [66]. A synthetic form of relaxin-3 named H3 showed a protective effect in a rat model of diabetic retinopathy and in hRMEC exposed to high glucose via a P2X7R-NLRP3-mediated mechanism. The synthetic compound MCC950, a sulfonylurea analog, also inhibited AGE-induced retinal damage, suppressed apoptosis, and normalized hRMECs migration. Combined application of H3 relaxin and MCC950 did not produce a synergistic effect, suggesting that the inhibitory activity occurred via a shared pathway, i.e., the P2X7R-NLRP3 axis [66]. As a further therapeutical approach, we may consider to exploit the reported inhibition of the P2X7R by the well-known anti-inflammatory agent colchicine [67]. Colchicine, a microtubule depolarizing drug with a potent anti-inflammatory activity, currently used to treat acute gout flares, familial Mediterranean fever (FMF), and acute pericarditis, disrupts the microtubule network and at the same time inhibits eATP-induced, P2X7R-dependent plasma membrane permeabilization to high MW dyes, ROS, nitric oxide (NO), and IL-1β release, without affecting ATP-evoked ion currents [67]. Thus, colchicine might be an additional low cost and widely available therapeutical option for the treatment of diabetic retinopathy.

## 5. Conclusions and Future Perspectives

Retinopathy is one of the most dreadful complications of diabetes. The role of inflammation and neoangiogenesis induced by chronically elevated levels of blood glucose is well documented and confirmed “ex adjuvantibus” by the efficacy of locally-administered anti-neoangiogenic drugs. The P2X7R emerged as a promising target for new pharmacological treatments because of its central role as an upstream modulator of inflammation, neoangiogenesis, and neural damage. Many small molecule drugs have been tested in vitro and in vivo pre-clinical models to block or down-modulate P2X7R with promising results. We are confident that purinergic signaling will provide new useful knowledge on the pathophysiological mechanisms underlying diabetic retinopathy and novel therapeutical options.

## Figures and Tables

**Figure 1 ijms-22-07110-f001:**
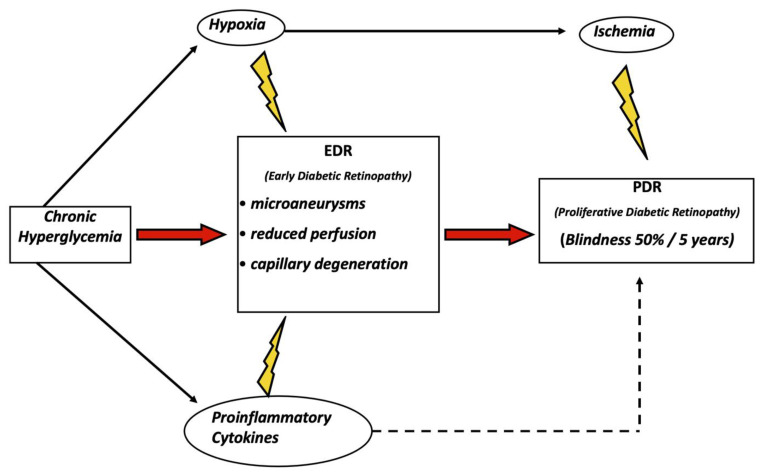
Schematic rendition of the main causative events in the pathogenesis of diabetic retinopathy.

**Figure 2 ijms-22-07110-f002:**
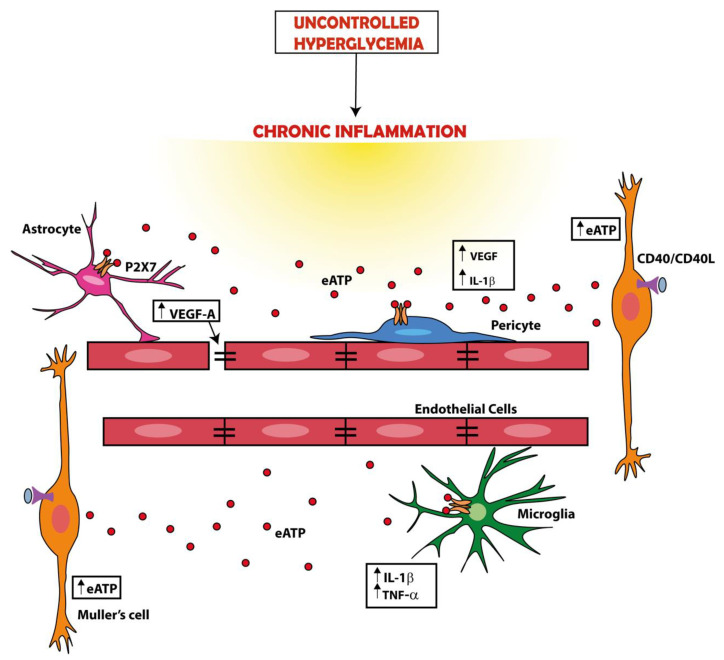
A dysregulation of the ATP-P2X7R axis is a main factor in the pathogenesis of diabetic retinopathy. Hyperglycemia is a cause of chronic inflammation, tissue damage, and cytokine (e.g., IL-1β or TNF-α) release. These events promote ATP release. For example, ATP release can be promoted by ligation of CD40 by CD40L on the plasma membrane of Müller cells or by the activation of the P2X7R itself. The increase in the extracellular ATP (eATP) concentration amplifies inflammation by activating the P2X7R expressed in the retina by several cell types (e.g., Müller cells, astrocytes, microglia, and pericytes), thus causing further release of cytokines and vascular endothelial growth factor (VEGF). VEGF in turn causes a breakdown of the blood–retinal barrier (BRB) promoting the formation of inflammatory exudate and inflammatory cell migration form circulation.

## Data Availability

Not applicable.

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
