# Peer review of "The P2X7 Receptor: A Promising Pharmacological Target in Diabetic Retinopathy"

_ijms, 2021, doi:10.3390/ijms22137110_

Round 1

Reviewer 1 Report

The authors summarize recent knowledge about diabetes-associated retinal degeneration,  and a possible role of P2X7 receptor in this disease. The manuscript  is well written, interesting and easy to read. I have only few comments and suggestions:

  1. List of abbreviations is absolutely missing, and since this manuscript contains many abbreviations, it is needed. Some abbreviations are used only 1 – 2 times, and thus can be omitted, for example “TJs” (line 275),”PDR” (line105) and “ESRD” (line 88) are used only 1x. Some abbreviations are explained several times, for example “ROS” is explained in line 223 and 312, and some are not explained, “PI3K/Akt pathway (line 314)”, for example.
  2. Line 264-265: “Many cell components of the retina, including neurons, glia and retinal pigmented epithelium, express the P2X7R....”The expression of P2X7 in neurons is questionable, and, according to my knowledge, it has not yet been documented. The authors should consider following change in this sentence: „Many cell components of the retina, including neuronal cells and retinal pigmented epithelium, express the P2X7R...”
  3. Paragraph 4 is very long and should be divided into several subsections.

Author Response

We wish to thank the Reviewers for their kind comments that we hopefully satisfied in the attached revised version.

  1. List of abbreviations is absolutely missing, and since this manuscript contains many abbreviations, it is needed. Some abbreviations are used only 1 – 2 times, and thus can be omitted, for example “TJs” (line 275),”PDR” (line105) and “ESRD” (line 88) are used only 1x. Some abbreviations are explained several times, for example “ROS” is explained in line 223 and 312, and some are not explained, “PI3K/Akt pathway (line 314)”, for example.

We are somewhat puzzled by this comment as Instructions to Authors for this Journal indicate that there should be no Abbreviation List, but rather abbreviations should be indicated when first encountered. We apologize for the missing explanation of some abbreviations, and for having replicated some more than once. These mistakes have now been amended.

  1. Line 264-265 (line 271-272 in the new version): “Many cell components of the retina, including neurons, glia and retinal pigmented epithelium, express the P2X7R....”The expression of P2X7 in neurons is questionable, and, according to my knowledge, it has not yet been documented. The authors should consider following change in this sentence: „Many cell components of the retina, including neuronal cells and retinal pigmented epithelium, express the P2X7R...”

We agree with the Reviewer that expression of the P2X7R by neurons is a contenctious issue. See for example the reviews by Peter Illes and the late Maria Teresa Miras-Portugal, respectively, reporting opposite views (Neuronal P2X7 Receptors Revisited: Do They Really Exist? Illes P, Khan TM, Rubini P. J Neurosci. 2017;37:7049-706) (Neuronal P2X7 Receptor: Involvement in Neuronal Physiology and Pathology. Miras-Portugal MT, Sebastián-Serrano Á, de Diego García L, Díaz-Hernández M. J Neurosci. 2017;37:7063-7072). Following the Reviewer’s suggestion we have changed the sentence at line 271-272 to: “Many cell components of the retina, including neuronal cells and retinal pigmented epithelium and possibly neuronal cells, express the P2X7R”.

  1. Paragraph 4 is very long and should be divided into several subsections.

We split Paragraph 4 in subsections.

Reviewer 2 Report

The review is quite timely and in general is very well written. It is interesting to put P2X7 R upstream of VEGFR and it is fitting. The 2 figures are fine.

The organization of the chapter needs to be reworked.  The title is The P2X7 receptor.. and yet there is no discussion of any purinoreceptor until 3.4.

Put section 3.4 after section 2.0.  The reviewer suggests giving  a description of what P2X7 is in general and how it responds to ATP (number of binding sites).

In the current section 3.4 note that  other variants than b as J and potentially k variants may be overexpressed in diabetes. Here there could be a description of what the response might be depending on transcriptional regulation. Is b always found in retinopathy and j or k in other tissues? 

section 2.-diabetes mellitus

The first sentence is simplistic and perhaps the CDC sentence could follow it. 

The second sentence from the end of the current section 2.0 should state ocular and not only retina as it is known that the cells in the cornea for example change, there is nerve degeneration which is extremely painful and there is a change in repair. While the retinopathy may lead more directly to blindness, the corneal pain leads to many requests for patient care. 

2.1 - Have the treatments altered the regulation of P2X7R? The chapter seems to be a list of events but they are not integrated.

Author Response

We wish to thank the Reviewers for their kind comments that we hopefully satisfied in the attached revised version.

1. The review is quite timely and in general is very well written. It is interesting to put P2X7 R upstream of VEGFR and it is fitting. The 2 figures are fine.

We thank the Reviewer for his/her detailed and helpful comments.

2. The organization of the chapter needs to be reworked.  The title is The P2X7 receptor.. and yet there is no discussion of any purinoreceptor until 3.4.

We respectfully disagree with this criticism. It is certainly true that the P2X7 receptor appears late in the Review, yet we reckon to be important that the reader, especially the lay reader, gets an overall picture of the disease (i.e. diabetes) before being introduced to the role that P2X7 might play in its pathogenesis.

3. Put section 3.4 after section 2.0.  The reviewer suggests giving  a description of what P2X7 is in general and how it responds to ATP (number of binding sites).

We are not sure to have fully understood this comment: section 2.0 deals with the pathophysiology of diabetes, while section 3.4 introduces purinergic signaling and P2X7, thus in our opinion it is very reasonable that section 3.4 follows sections 3.1 (neovascularization, 3.2 (inflammation) and 3.3 (neuronal damage). As to a description of the P2X7R, a few sentences were added (lines 218-223).

4. In the current section 3.4 note that  other variants than b as J and potentially k variants may be overexpressed in diabetes. Here there could be a description of what the response might be depending on transcriptional regulation. Is b always found in retinopathy and j or k in other tissues?

To our knowledge, there is no study reporting selective expression or overexpression of any P2X7 isoform in diabetes, thus we are unable to  answer this question.

5. section 2. diabetes mellitus The first sentence is simplistic and perhaps the CDC sentence could follow it. The second sentence from the end of the current section 2.0 should state ocular and not only retina as it is known that the cells in the cornea for example change, there is nerve degeneration which is extremely painful and there is a change in repair. While the retinopathy may lead more directly to blindness, the corneal pain leads to many requests for patient care.

We agree the first sentence of paragraph 2.0 is simplistic, but it was meant to be a general definition of a very complex and multifaceted disease that we further described later. “Ocular” replaced “retina” at line 87.

6. 2.1 - Have the treatments altered the regulation of P2X7R? The chapter seems to be a list of events but they are not integrated.

There is no in depth investigation in eye diseases of the effects of treatments on P2X7R expression. We apologize for loose cohesion of this chapter.

Round 2

Reviewer 2 Report

In a search by the reviewer there is documentation of the j variant that is elevated in human diabetic epithelium. 

The reviewer asks that the authors at the least state that there are variants and its not known if there are differences in the retina.

P2X7 exists as a trimer and that should be stated under the general (3.4). There are 3-5 binding sites found in the literature and it appears to be species specific.

If the organization is not going to be altered,  the title should not start with P2X7 as it is not mentioned until 3.4. It may be easier to alter the title.

Author Response

Please, see attached pdf.
